# Protocol of economic evaluation and equity impact analysis of mHealth and community groups for prevention and control of diabetes in rural Bangladesh in a three-arm cluster randomised controlled trial

Hassan Haghparast-Bidgoli,[1] Sanjit Kumar Shaha,[2] Abdul Kuddus,[2]
Md Alimul Reza Chowdhury,[2] Hannah Jennings,[1] Naveed Ahmed,[2]
Joanna Morrison,[1] Kohenour Akter,[2] Badrun Nahar,[2] Tasmin Nahar,[2] Carina King,[1]
Jolene Skordis-Worrall,[1] Neha Batura,[1] Jahangir A Khan,[3] Anthony Mansaray,[4]
Rachael Hunter,[5] A K Azad Khan,[2] Anthony Costello,[1,6] Kishwar Azad,[2]
Edward Fottrell[1]

For numbered affiliations see end of article.

**Correspondence to**
Dr Hassan Haghparast-Bidgoli;
h.haghparast-bidgoli@ucl.ac.uk

## ABSTRACT

**Introduction** Type 2 diabetes mellitus (T2DM) is one of the leading causes of death and disability worldwide, generating substantial economic burden for people with diabetes and their families, and to health systems and national economies. Bangladesh has one of the largest numbers of adults with diabetes in the South Asian region. This paper describes the planned economic evaluation of a three-arm cluster randomised control trial of mHealth and community mobilisation interventions to prevent and control T2DM and non-communicable diseases' risk factors in rural Bangladesh (D-Magic trial).

**Methods and analysis** The economic evaluation will be conducted as a within-trial analysis to evaluate the incremental costs and health outcomes of mHealth and community mobilisation interventions compared with the status quo. The analyses will be conducted from a societal perspective, assessing the economic impact for all parties affected by the interventions, including implementing agencies (programme costs), healthcare providers, and participants and their households. Incremental cost-effectiveness ratios (ICERs) will be calculated in terms of cost per case of intermediate hyperglycaemia and T2DM prevented and cost per case of diabetes prevented among individuals with intermediate hyperglycaemia at baseline and cost per mm Hg reduction in systolic blood pressure. In addition to ICERs, the economic evaluation will be presented as a cost–consequence analysis where the incremental costs and all statistically significant outcomes will be listed separately. Robustness of the results will be assessed through sensitivity analyses. In addition, an analysis of equity impact of the interventions will be conducted.

**Ethics and dissemination** The approval to conduct the study was obtained by the University College London Research Ethics Committee (4766/002) and by the

### Strengths and limitations of this study

► This protocol paper reports planned data collection and analyses alongside a complex public health trial to ensure transparency.
► This protocol can assist in designing mHealth and community mobilisation through participatory learning and action approach interventions for the prevention of diabetes and mitigation of non-communicable diseases risk factors.
► The protocol, and planned analysis and reporting follow recommended guidelines to design and report economic evaluations.
► An equity impact analysis and contingent valuation study will be conducted alongside the study.
► The study is powered to assess the cost-effectiveness of each intervention against control only. The study is not powered to test the differences between the mHealth and community mobilisation interventions. Therefore, it will be possible to estimate incremental cost and effect of each intervention compared with the control only.

Ethical Review Committee of the Diabetic Association of Bangladesh (BADAS-ERC/EC/t5100246). The findings of this study will be disseminated through different means within academia and the wider policy sphere.
**Trial registration number** ISRCTN41083256; Pre-results.

## BACKGROUND
### Burden of diabetes mellitus
Diabetes mellitus, mainly type 2 (T2DM), is one of the leading causes of death and disability worldwide. It is estimated that

around 415 million people worldwide or 9% of adults aged 20–79 have diabetes, with about 75% of people living with diabetes residing in low-income and middle-income countries (LMICs).[1] It is predicted that by 2040, 1 in 10 adults will have diabetes, unless preventive efforts are undertaken.[1] Diabetes, if not managed properly, can lead to several complications, such as heart attack, kidney failure, leg amputation, vision loss and several other long-term consequences, that impact significantly on quality of life and cause premature death.[1 2] Diabetes and its complications create a substantial economic burden for people with diabetes and their families, and for health systems and national economies through direct medical costs and productivity loss.[2] It is reported that in many countries between 5% and 20% of total health expenditure are spent on diabetes.[1] This is in addition to the large financial burden on individuals and their families due to the cost of seeking care.

Bangladesh has the second largest number of adults with diabetes in the South Asian region, with around 7 million adults aged 20–79 years with diabetes.[1] The prevalence of diabetes among adults (20–79 years old) in Bangladesh is estimated to be around 8.5%,[1 2] with a three to fourfold increase since the 1990s.[3 4] It is predicted that prevalence of diabetes in Bangladesh will reach 23.6% in men and 33.5% in women by 2030, unless preventive efforts are undertaken.[3] The increasingly high incidence of diabetes in Bangladesh has had a significant economic burden, in particular for people with diabetes, their families and the country's healthcare system. It is estimated that the annual health expenditure for diabetes is around US$218 million, where most of the costs born by the families.[5]

Underlying the increasing prevalence of diabetes, globally and in Bangladesh, are complex genetic, environmental and lifestyle factors, including changes in dietary habits and increases in risk factors like smoking and physical inactivity.[2] Effective interventions are available to prevent T2DM and to prevent the complications and premature death from diabetes.[2] The evidence from LMICs shows that lifestyle and other non-pharmacological interventions can prevent and delay the onset of T2DM and its complications.[6] However, there is a lack of cost-effective programmes designed specifically for Bangladeshi populations, taking into account their contextual needs and resources.

## The Bangladesh D-Magic trial

The Bangladesh D-magic is a three-arm cluster randomised control trial (cRCT) of mHealth and community mobilisation interventions, conducted in four rural upazilas in Faridpur district, Bangladesh. The trial aims to assess the effectiveness and cost-effectiveness of the interventions in the prevention and control of T2DM and non-communicable disease (NCD) risk factors in rural Bangladesh. In the D-magic trial, the clusters (96 villages) were randomised to receive either the mHealth intervention, the community mobilisation intervention or be in the control arm (32 in each). In the mHealth intervention, individuals receive voice messages about prevention and control of NCD risk factors and T2DM on their mobile phone. In the community mobilisation intervention, a trained facilitator initiates a series of diabetes and NCD risk factor focused monthly group meetings for men and women, working through a participatory learning and action (PLA) cycle by which group members themselves identify, prioritise and tackle problems associated with T2DM and its risk factors. In addition, all study areas (both intervention and control) clusters receive health system strengthening (HSS) activities, which include the training/refresher training of healthcare workers working in the community and health facilities, in the prevention, diagnosis and treatment of T2DM, as well as the development of essential equipment inventories.

Details of the D-Magic trial are described elsewhere.[7] This protocol paper aims to fully describe the methodology for the economic evaluation of the trial.

## Economic evaluations of community and mHealth interventions for prevention and control of diabetes

There is evidence that mHealth programmes can have a positive impact on behavioural change and prevention and control of diabetes and NCDs in high-risk populations.[8–11] However, there is little information on the cost and cost-effectiveness of mHealth interventions for the prevention and control of NCDs.[12–14] Two recent systematic reviews of the economic evidence of mHealth[13] and mHealth for diabetes prevention and control[14] have shown that there are a handful of NCD and diabetes interventions that have reported cost and cost-effectiveness evidence. Nearly, all of these studies have been conducted in high-income settings. The majority of these studies report that mHealth interventions are cost-effective or cost saving, though the quality of reported evidence was not satisfactory in some of cases.[13 14]

Similarly, although there is some evidence on effectiveness of community-based interventions in the management of T2DM in low-income settings,[15–19] there is little evidence on how cost-effective these interventions might be.[14] A recent review[14] has identified 10 community-based interventions on preventing and controlling diabetes. These interventions, which are largely implemented in high-income settings, have reported that community-based interventions are cost-effective or cost-saving approaches in the management of T2DM.[14]

The current study will be the first to assess the cost-effectiveness of community mobilisation through PLA in the prevention of T2DM. This approach has previously been shown to be highly cost-effective in improving maternal and newborn health.[20 21] This study will also contribute to the evidence on cost-effectiveness of mHealth interventions for preventing T2DM in LMIC settings.

## Aim and objectives

The D-Magic economic evaluation aims to measure the cost-effectiveness of mHealth and participatory

community group interventions to prevent and to control T2DM and NCD risk factors in rural Bangladesh from a societal perspective.

The specific objectives of the D-Magic economic evaluation are:

1. To estimate the costs of setting up and implementing the mHealth and participatory community group interventions as well as HSS activities.
2. To calculate the costs to the healthcare system, of increased care-seeking (ie, diagnosis and treatment) for T2DM and other NCDs, as a result of the D-Magic interventions.
3. To measure costs associated to the intervention participants and their households of changes in diabetes or other NCD-related diagnosis and management care-seeking costs as well as any costs associate with changes in diet and other lifestyle behaviours, as a result of the D-Magic interventions.
4. To present the incremental costs and outcomes of the interventions as a cost–consequence analysis.
5. To calculate the incremental cost-effectiveness of the mHealth and community mobilisation interventions combined with HSS activities, as compared with HSS activities alone, where all new HSS activities are delivered in addition to the existing government programmes.

In addition to the above-mentioned objectives, the equity impact of the mHealth and community mobilisation interventions will be assessed.

## METHODS
### Study setting and population

The study setting for the D-Magic trial is Faridpur district, located South of Dhaka. The district has a population of over 1.7 million people in a high-density area of just over 2000 km$^2$. It is primarily an agricultural economy, with the main crops being jute and rice. Like in the rest of Bangladesh, healthcare is provided at three levels: primary care is provided at Community Clinics and at Union Health and Family Welfare Centres; secondary level care providers (both inpatient and outpatient services) are subdistrict (upazila) health complexes and hospitals; and tertiary care is provided at district hospitals and medical college hospitals.[22] Private and informal service providers are also present in the area, with the informal sector being the main provider in rural areas,[22] Faridpur district included. Inadequate and inequitable access to services, shortages of skilled healthcare providers, short supplies of medicines and poor quality and low responsiveness of services are the main challenges faced by the Bangladesh healthcare system,[22] and remains a challenge in Faridpur district too.

The study population for the D-Magic study is men and non-pregnant women who are aged 30 years or more and permanent residents (ie, lived there for more than 6 month) of 96 villages (clusters) in four rural upazilas—Nagarkanda, Boalmari, Saltha and Madhukhali—in Faridpur district.

### Trial design

The D-Magic trial is a three-arm cRCT which is implemented in four rural upazilas in Faridpur district, Bangladesh. In D-Magic trial, 96 clusters (villages) were randomised to receive either the mHealth intervention, the community mobilisation intervention or be in the control arm. All three arms receive a number of HSS activities. The interventions implementation was started in July 2016 and was completed by end of December 2017 and all data collection was completed in June 2018.

Detailed information on the randomisation and participants recruitment process, trial timeline, and a full description of the interventions is presented elsewhere.[7] A brief description of the mHealth, community mobilisation and HSS interventions are presented in the following sections.

### mHealth intervention

The mHealth intervention involves free of charge voice messages about the prevention and control of T2DM and NCD risk factors sent two times per week to the individual's mobile phone during a 14-month period. The intervention and the messages' content were developed based on findings from baseline formative research in the study area and application of the behavioural change theories such as the Capability, Opportunity, Motivation, Behaviour (COM-B) model for understanding behaviour[23] and the Theoretical Domains Framework (TDF) to encourage change.[24 25] The mHealth intervention is available to all individuals who have access to a mobile phone and registered to receive the messages by providing their mobile number to the intervention community recruiters.

### Community mobilisation intervention

The community mobilisation intervention involves initiation and facilitation of separate male and female participatory groups, with approximately 20 members in each. The intervention is an adaptation of a participatory women's groups intervention implemented in South Asia (including Bangladesh) and Sub-Saharan African settings and shown to be effective and cost-effective at reducing neonatal mortality.[20 26] The groups progress through a series of 18 monthly meetings following the four phases of PLA. During phase 1, participants identify and prioritise factors that affect their health, particularly those increasing their risk of developing or failing to manage T2DM; in phase 2, the participants and their community come up with feasible strategies that can be implemented to address the problems identified in phase 1; during phase 3, they implement these strategies; in phase 4, they evaluate the strategies they have implemented. The groups are run by salaried facilitators, who are recruited from the study areas and have a minimum of higher secondary school education. The facilitators undertook 1-week training on group facilitation and basic health messages related to NCD prevention and control, in particular T2DM. They also have

received refresher training during the course of the interventions.

## Health systems strengthening

A series of HSS activities are carried out in all study areas (both intervention and control clusters). These activities, which are tailored according to the mapping of healthcare providers in the project area in the formative phase of the project, included the training of mainly informal healthcare workers in the community and distributing educational materials among formal and informal providers in prevention, diagnosis and treatment of T2DM, as well as the development of essential equipment inventories.

## Measurement of health outcomes/effectiveness

The D-Magic trial will test the effect of the community mobilisation intervention and mHealth intervention relative to the control and does not directly compare the effects of each intervention relative to the other. As this is a cluster randomised trial, the outcomes will be measured among individuals (permanent residents) who live in the intervention clusters, irrespective of whether they took part in groups or received mHealth messages. Analysis of the outcomes will be by intention to treat at the individual and cluster level as appropriate. Moreover, participants with missing data on the primary outcomes will be excluded from primary outcome analysis.[7]

### Primary outcome

The D-Magic trial has two primary outcomes: combined prevalence of intermediate hyperglycaemia (ie, impaired fasting glucose or impaired glucose tolerance) and T2DM, and cumulative 2-year incidence of T2DM among individuals identified with intermediate hyperglycaemia at the start of the trial.[7] Using the prevalence data, the number of intermediate hyperglycaemia and diabetes mellitus cases prevented, and the number of diabetes cases prevented among individuals with intermediate hyperglycaemia at baseline will be calculated as the difference between the expected and the actual number of cases using the adjusted OR relative to the control population.

### Secondary outcomes

The trial has a number of secondary outcomes including diastolic and systolic blood pressure (BP), prevalence of hypertension, body mass index, prevalence of overweight and obesity, prevalence of abdominal obesity, health-related quality of life, and psychological distress among self-reported diabetics.[7]

Comparison will be made between the interventions (mHealth and community mobilisation) and control (HSS activities only) to estimate incremental cost-effectiveness ratios (ICERs) for the primary outcomes; in terms of cost per case of intermediate hyperglycaemia and diabetes mellitus prevented and cost per case of diabetes prevented among individuals with intermediate hyperglycaemia at baseline, and for some of the secondary outcomes such as cost per mm Hg reduction in systolic BP. ICER will be conducted if a significant impact on the

outcomes is observed. Moreover, all costs and (statistically significant) outcomes, both primary and secondary, will be presented separately in a cost–consequence analysis.

## Willingness to pay/contingent valuation study

We will conduct a willingness to pay (WTP) study in order to elicit maximum monthly amount of money each participant or household would be willing to pay if an mHealth service on diabetes and NCD risk factors' prevention and management (ie, weekly voice messages) was available.

WTP studies are widely used in order to elicit monetary value of a service or good not available in the market.[27–30] In the Bangladesh context, Shariful Islam *et al* in a recent study estimated WTP of patients with T2DM for receiving messages for increasing adherence to treatment.[31] Similar to Shariful Islam *et al*, we will use an open-ended contingent valuation (CV) method,[27 30 32] asking participants through an open-ended question how much they would be willing to pay monthly to receive voice messages related to diabetes and NCD risk factors' prevention and management, if such a service was available. Open-ended CV is a more flexible approach and avoids starting point bias or range bias introduced by other methods such bidding game and payment cards.[27] The theoretical framework defined by O'Brien and Gafni[28] for CV studies will be used to design the study.

WTP questions will be asked from all the participants at the end-line impact evaluation survey. WTP values will be compared across different groups, for example, based on exposure to the interventions (ie, among those exposed to mHealth or exposed to participatory groups and participants who have not been exposed to either of these interventions) or health condition (people diagnosed with diabetes or other NCDs and others). In addition, for each participant, detailed individual-level and household-level socioeconomic characteristics will be collected to examine the extent to which WTP values will vary by socioeconomic status of participants.

## Equity impact of the D-Magic interventions

In addition to measuring efficiency/cost-effectiveness of the D-Magic interventions, an analysis of equity impact of the interventions will be conducted to assess whether impacts/gains from the intervention are equitably shared among the target population. This will be done through subgroup analyses of the primary and secondary outcomes based on the socioeconomic status of the target population.

## Identification, measurement and valuation of resource use

The cost-effectiveness and cost–consequence of the D-Magic interventions will be measured from a societal perspective[33 34]; measuring the economic impact for all parties affected by the interventions, including implementing agency (or programme costs), public healthcare providers (at both local and national levels) and users, who are the intervention participants and their households. The following sections provide a detailed description

**Table 1** Overview of resource use and costs measures included in the economic evaluation of the D-Magic interventions

| Perspective/cost category | Type of costs | Description | Sources | Sample size |
|---|---|---|---|---|
| **Provider** | | | | |
| Programme/ implementing agency | Direct | Costs of implementing mHealth, community mobilisation and HSS interventions. | 1. Project accounts of the implementing agencies. 2. Interviews with the project staff. | NA |
| | Indirect | The opportunity cost of volunteer experts attended the mHealth design meetings, donated items, etc. | 1. Project records on numbers of meeting, attendants, etc. 2. Published reports on local wage information based on skill category. 3. Field offices' inventory information. | 1. All meetings held and number of people attended in the meetings. 2. The list of all equipment in the field offices. |
| Public healthcare providers | Direct | Changes in utilisation of T2DM and NCD-related services at the public health facilities in the study area. | 1. Detailed audit and costing study of the health facilities. 2. Baseline and end-line cross-sectional surveys (for information on changes in costs of health care seeking). | 1. Random sample of health facilities at different levels in both intervention and control areas. 2. All participants in the study. |
| | Indirect | The opportunity cost of the time spent by the healthcare providers attending HSS meetings. | 1. Project records on numbers of meeting, attendants, etc. 2. Published reports on local wage information based on skill category. | All meetings held and number of people attended in the meetings. |
| Participants/ households | Direct | Household expenditure on food and non-food. | Household consumption expenditure survey. | A random sample of 300 households in the study area |
| | | Costs of health care seeking for the participants and their households. | Baseline and end-line cross-sectional surveys. | All participants in the study. |
| | Indirect | Opportunity cost of participation in the groups. | Group participants survey. | A random sample of 312 group participants (both male and female). |

HSS, health system strengthening; NA, not applicable; NCD, non-communicable diseases; T2DM, type 2 diabetes mellitus.

of the proposed methods for measuring and valuing programme costs, healthcare provider costs and participants/household costs. Programme costs include those incurred by the implementing agency or programme provider, that is, Diabetic Association of Bangladesh. The healthcare provider costs are those incurred by the government health facilities in the study area including community clinics, Health and Family Welfare centres and Upazila Health Complex/hospitals. The household or user costs include those incurred by participants and their households. Table 1 provides an overview of the financial and economic costs to be employed in the economic evaluation of the D-Magic interventions.

## Programme-related costs

Direct and indirect costs of designing and implementing mHealth, community mobilisation and HSS interventions will be estimated using a combination of activity-based costing[35] and ingredients approach.[36]

### Financial or expenditure data

Programme costs are mainly financial or accounting costs, which are collected prospectively from the project accounts or expenditure records of the implementing partner and entered (generally, on an annual basis) to an MS Excel data capture tool. The tool contains different sections/worksheets that will allow the categorisation of costs into line items (ie, staff, materials, capital and joint costs), start up and implementation costs of the interventions and costs associated with the different programme components, that is, mHealth, community mobilisation, HSS, and monitoring and evaluation. Key informant interviews with project leads and monthly/quarterly staff time sheets will be used to allocate joint costs between the programme components. The summary worksheets in the cost data capture tool present the costs by programme component (eg, mHealth, community mobilisation and HSS), summarise the total cost data per intervention, allows effect data to be entered and estimates the cost-effectiveness results.

## Donated items and opportunity costs

Some items, such as donated items and volunteer time, are not captured in the accounting system and need to be converted to economic costs using their market value and then entered into the data capture tool.[37–39] Potential donated items are equipment donated by the implementing agency (ie, purchased by previous projects and used in the D-Magic project). The donated items will be identified through key informant interviews with the project leads.

The majority of volunteer time is related to designing messages for the mHealth intervention, where several meetings were held with experts who were volunteers. Detailed information regarding these meetings, including the number of meetings, their duration and the participants, is being documented by the project. The opportunity cost of the time invested by the experts will be measured as a proportion of their salary or a salary equivalent using published national/local wage rate reports.

### Public healthcare providers costs

The D-Magic project is likely (at least in the short term) to increase seeking care for diabetes, other NCDs and NCD risk factors such as hypertension, particularly demand for services such as testing and treatment for hypertension, diabetes and prediabetes, or seeking advice or treatment for weight control. In addition, there is a time (opportunity) cost of direct involvement in the HSS activities for the healthcare providers (table 1).

### Cost of changes in demand for services

The costs to public healthcare providers in the project area due to increased (or any changes in) demand for their services will be estimated. A mapping of the healthcare providers in the study area has been completed and 20 functional governmental healthcare facilities, at different levels, have been identified. These facilities included 14 Community Clinics, 3 Health and Family Welfare centres and 3 Upazila Health Complex/Hospitals. A sample of 11 health facilities including 8 community clinics and 2 Health and Family Welfare centres (with equal numbers in control and intervention clusters), as well as 1 Upazila Health Complex/Hospital (which cover both control and intervention areas) was selected for baseline audits of diabetes and NCD services, estimating their resource utilisations and unit costs. Similar audits and cost data collection will be conducted for the same facilities post intervention in order to assess the changes in NCD service utilisation attributable to the D-Magic interventions. This data will be complemented by health-seeking behaviour information collected from the study participants in control and interventions clusters at the D-Magic end-line impact evaluation survey. Differences in service utilisation between intervention and control areas will be attributed to the D-Magic interventions.

A simple audit and cost-capture tool was developed for facility data collection and piloted with facilities at different levels. Data from the cost-capture tool will be complemented by the existing data from the facility reports and published data. Costs of services provided by the facilities will be estimated using a step-down approach.[40]

Any change in demand for services provided in the facilities other than those mentioned above and the services not covered by them, in intervention areas compared with the control, will be identified during the trial's routine monitoring and end-line impact evaluation survey. Any cost of that change in demand will be calculated using published data on the unit costs of those services.

### Opportunity cost of HSS activities

Moreover, as discussed earlier, HSS activities include several training sessions for healthcare providers in prevention, diagnosis and treatment of T2DM. Information on the number of meetings, their duration and participation is being documented by the project. The opportunity cost of the time spent by the providers will be measured as a proportion of their salary for formal providers or as a salary equivalent, for informal providers.

### Participants and their household costs

D-Magic interventions may influence participants and their households' costs in a number of ways. These include changes in health-seeking behaviour, and changes in household lifestyles that might affect food and non-food consumption patterns and spending as well as time spent engaging in physical activities. It also includes the time (opportunity) cost of participation in the PLA group meetings and participating in the actions taken by the groups (table 1).

### Healthcare-seeking costs

D-Magic may increase the participants and their households' seeking advice and care from both formal and informal providers for testing and treatment for hypertension, diabetes, prediabetes and other NCDs. Information on costs of care-seeking is collected from all the participants recruited in the project, at the baseline and end-line evaluation surveys. This information will be complemented by the data collected from the household consumption expenditure survey. The difference in spending for the participants and their households will be calculated and compared between intervention and control areas.

### Changes in household food and non-food expenditure

Changes in food and non-food expenditure will be captured in a comprehensive household consumption and expenditure survey. The survey will be conducted on a random subsample of 300 households (100 per trial arm) at the end of intervention period. The changes in the expenditure will be compared between interventions and control areas.

### Opportunity cost of participation in the interventions

Participating in the PLA group meetings incur some costs to the participants and their family. These costs include

the direct costs (eg, cost of getting to the group) and time cost of group participation (eg, travel time and time spent in the group) and participating in the actions taken by the group, or changing health and lifestyle behaviours. Information on the potential direct and time costs will be collected through a subsample survey of 312 group participants (both male and female), randomly selected. Sample size for the survey was primarily calculated to give sufficiently accurate estimate of group participants' characteristics.

## Cost-effectiveness and cost–consequence analyses

Economic evaluation will be conducted as a within-trial analysis using the intention-to-treat results, and will be presented in terms of ICERs, calculated as the difference in total costs of mHealth and community mobilisation interventions (plus HSS activities) versus HSS activities only (or control), divided by the difference in mean effects of each interventions versus control.[41 42] As mentioned previously, ICERs will be evaluated in terms of cost per case of intermediate hyperglycaemia and T2DM prevented and cost per case of T2DM prevented among individuals with intermediate hyperglycaemia at baseline, and for some of the secondary outcomes such as cost per mm Hg reduction in systolic BP. In addition to ICERs, the economic evaluation will be presented as a cost–consequence analysis where the incremental costs and all statistically significant outcomes will be listed separately, allowing policy-makers to compare the costs and all impacts/gains of the D-Magic interventions. Cost–consequence analysis has been recommended for complex public health interventions, such as D-Magic, that have multiple health and non-health impacts, which are difficult to measure in a common outcome unit.[42 43]

All costs will be presented in 2017 prices in Bangladeshi Taka and International Dollars (INT\$). All costs will be adjusted for inflation using the Bangladeshi Consumer Price Index and will be converted to 2017 INT\$ using the purchasing power parity conversion factor for Bangladesh. Moreover, both costs and outcomes will be discounted using a standard discount rate of 3%, as recommended by WHO-CHOICE[44] and the Gates/IDSi Reference Case for Economic Evaluation.[45] The impact of uncertainty in key parameters on the cost-effectiveness results will be assessed through a series of deterministic and probabilistic sensitivity analyses. Reporting of the study design, analytical methods and findings will follow the Consolidated Health Economic Evaluation Reporting Standards (CHEERS) statement.[46] The D-Magic interventions will be judged to be cost-effective and affordable (though, indirectly) against the WHO-CHOICE recommendation,[44] as well as recently developed cost-effectiveness thresholds.[47]

The possibility of conducting an extended cost-effectiveness analysis[48] alongside modelling national scale up of the D-Magic interventions will be explored. Moreover, we will explore the possibility of running a long-term cost-effectiveness analysis using decision analytical modelling based on the relevant outcomes such as systolic BP, body mass index (if statistically significant) to predict future economic impacts from implementing the D-Magic interventions on the target population.

## Strengths and limitations of the study

In order to increase transparency and minimise bias, publication and peer review of economic evaluation protocols is encouraged. This study reports planned data collection and analyses alongside a complex public health trial. The study will contribute to the scarce cost-effectiveness evidence on mHealth and community mobilisation interventions for preventing diabetes and NCDs.

Furthermore, adopting a cost–consequence analysis approach makes it possible to report all health and non-health impacts of the D-Magic interventions, in addition to ICERs, which can assist policy-makers to make informed decisions in designing or implementing similar complex interventions. Incorporating equity impact analysis and CV are other strengths of this study, which provide useful information for future scale-up of the interventions.

The study design has one limitation. The D-Magic trial is not powered to test the differences between the mHealth and community mobilisation interventions due to the large sample size required and the resources available for the trial. However, the possibility of a direct comparison between the two interventions will be explored.

## Patient and public involvement

Patients and public were not involved in the process of this study. Patients and public will be informed of the study results via peer-reviewed journals, conference and local dissemination meetings.

## Dissemination

The findings of this study will be disseminated through different means within academia and the wider policy sphere.

**Author affiliations**
[1]Institute for Global Health, University College London, London, UK
[2]Diabetic Association of Bangladesh, Dhaka, Dhaka District, Bangladesh
[3]Liverpool School of Tropical Medicine, Liverpool, UK
[4]ICAP Sierra Leone, Freetown, Sierra Leone
[5]Institute of Epidemiology & Health, University College London, London, UK
[6]World Health Organization, Geneva, Switzerland

**Contributors** HH-B, EF, AK, HJ, JM, CK, JS-W, NB, JAK, RH, AKAK, AC and KA contributed to the study design. HH-B, EF and CK contributed to the statistical analyses. SKS, MARC, NA, KA, BN and TN contributed to data acquisition. AM contributed to the literature review and drafting the paper. HHB was responsible for the initial drafting of this manuscript. All authors contributed to the review of this manuscript and provided comments. All authors read and approved the final manuscript.

**Funding** The trial is funded by the Medical Research Council UK (MR/M016501/1) under the Global Alliance for Chronic Diseases (GACD) Diabetes Programme.

**Competing interests** None declared.

**Patient consent** Not required.

**Ethics approval** The D-Magic trial has been approved by the University College London Research Ethics Committee (4766/002) and by the Ethical Review Committee of the Diabetic Association of Bangladesh (BADAS-ERC/EC/t5100246).

**Provenance and peer review** Not commissioned; externally peer reviewed.

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
