## [Reviewer comments · BMJ Open]

ARTICLE DETAILS

TITLE (PROVISIONAL)	Protocol of economic evaluation and equity impact of m-Health and community groups for prevention and control of diabetes in rural Bangladesh in a three-arm cluster randomised controlled trial
AUTHORS	Haghparast-Bidgoli, Hassan; Shaha, Sanjit; Kuddus, Abdul; Chowdhury, Md.Alimul Reza; Jennings, Hannah; Ahmed, Naveed; Morrison, Joanna; Akter, Kohenour; Nahar, Badrun; Nahar, Tasmin; King, C.; Skordis-Worrall, Jolene; Batura, Neha; Khan, Jahangir; Mansaray, Anthony; Hunter, Rachael; Azad Khan, A.K.; Costello, Anthony; Azad, Kishwar; Fottrell, Edward

VERSION 1 – REVIEW

REVIEWER	Samantha Paige University of Florida, USA
REVIEW RETURNED	13-Feb-2018

GENERAL COMMENTS	This submission presents a protocol to test the effectiveness of a diabetes mHealth and community-based intervention in rural Bangladesh. Authors describe the mHealth app, community program, and control group. The mHealth app and community program will be compared to the control group, but not to each other. Primary and secondary outcomes of the intervention is described, as well as the cost-effectiveness and cost-consequence variables. Authors describe that mHealth is increasingly valued as a low-cost public health tool, yet there is limited evidence for their efficacy (both improving diabetes outcomes and cost-related) in low income populations. The submission is valuable and of interest to transdisciplinary teams conducting international research on diabetes, as it presents a theoretically-driven protocol to implement and evaluate innovative methods to alleviate the burden of diabetes. Comments: 1) The description of each arm of the intervention was clearly described. Table 1 clearly outlined the economic evaluation plan, which was nicely supplemented in the text. 2) In the introduction, the authors describe the epidemiological data surrounding diabetes in Bangladesh to form a strong argument that this is a significant health problem. Despite stating that there is a substantial economic burden of diabetes, a thorough description of the economic burden in Bangladesh is limited. Also, much of the protocol focuses on the intervention's economic outcomes associated with the patient, healthcare provider, and system. Highlighting the current economic state (and economic burden of diabetes) of each group would help put the intervention design and analyses into perspective.
---

	3) In the methods (pg. 6; lines 55-56), "The interventions will be completed by the end of December and all data collection will be ongoing until May 2018." Please clarify December of which year. 4) Please justify why mHealth app intervention and community-based intervention are compared to the control group, but not compared to one another. 5) A section should highlight potential limitations, and potentially challenges and attempts to overcome them.
--	---

REVIEWER	Sheyu Li Department of Endocrinology and Metabolism, West China Hospital, Sichuan University
REVIEW RETURNED	02-Apr-2018

GENERAL COMMENTS	This is a cost-effective analysis of m-health for diabetes prevention and control based on a cluster-randomized trial. The study was well designed and the protocol was well presented. I have only some minor concerns before its publication.  1. How many people use smart mobile phones in rural Bangladesh? Is there any difference between the users and non-users? Will the equipment facility be concerned in the study? 2. I did not find QALY in the outcomes. How do the authors think about QALY? 3. Some critical information may be re-stated in the current protocol, such as the brief timeline of the trial (eg. the time when the first cluster/patient recruited) and the strategy of informed consent (or exempted). 4. I do suggest a brief discussion added following the methodology section. Some information could be discussed, such as the clinical and public expectation of the study as well as its expected strength and limitations.
---

VERSION 1 – AUTHOR RESPONSE

Reviewer: 1
Reviewer Name: Samantha Paige
Institution and Country: University of Florida, USA
Please state any competing interests: None.

Please leave your comments for the authors below

This submission presents a protocol to test the effectiveness of a diabetes mHealth and community-based intervention in rural Bangladesh. Authors describe the mHealth app, community program, and control group. The mHealth app and community program will be compared to the control group, but not to each other. Primary and secondary outcomes of the intervention is described, as well as the cost-effectiveness and cost-consequence variables. Authors describe that mHealth is increasingly valued as a low-cost public health tool, yet there is limited evidence for their efficacy (both improving diabetes outcomes and cost-related) in low income populations. The submission is valuable and of interest to transdisciplinary teams conducting international research on diabetes, as it presents a

theoretically-driven protocol to implement and evaluate innovative methods to alleviate the burden of diabetes.

Reply: We thank the reviewer for their kind appraisal of our work.

Comments:

1) The description of each arm of the intervention was clearly described. Table 1 clearly outlined the economic evaluation plan, which was nicely supplemented in the text.

Reply: We thank the reviewer for their kind appraisal of our work.

2) In the introduction, the authors describe the epidemiological data surrounding diabetes in Bangladesh to form a strong argument that this is a significant health problem. Despite stating that there is a substantial economic burden of diabetes, a thorough description of the economic burden in Bangladesh is limited. Also, much of the protocol focuses on the intervention's economic outcomes associated with the patient, healthcare provider, and system. Highlighting the current economic state (and economic burden of diabetes) of each group would help put the intervention design and analyses into perspective.

Reply: We agree with the reviewer that there is limited description on the economic burden of diabetes in Bangladesh. We have now added few sentences in the introduction explaining the financial burden of the diabetes, globally and in Bangladesh (page 4, second and third paragraphs).

3) In the methods (pg. 6; lines 55-56), "The interventions will be completed by the end of December and all data collection will be ongoing until May 2018." Please clarify December of which year.

Reply: We thank the reviewer for spotting this. We have now clarified the year (2017) in the text (page 6, last paragraph).

4) Please justify why mHealth app intervention and community-based intervention are compared to the control group, but not compared to one another.

Reply: We thank the reviewer for the comment. The D-Magic trial is not powered to test the differences between mHealth and community mobilisation interventions due to the large sample size required and the resources available for this trial. We have now highlighted this in the strengths and limitation section of the paper (Page 13, last paragraph)

5) A section should highlight potential limitations, and potentially challenges and attempts to overcome them.

Reply: We thank the reviewer for the comment. We have now added a section at the end of the discussion focusing on the strengths and limitations of the study on page 13.

Reviewer: 2

Reviewer Name: Sheyu Li

Institution and Country: 1 West China Hospital, Sichuan University, China; 2 Ninewells Hospital, University of Dundee, Scotland, UK.

Please state any competing interests: None to declare.

Please leave your comments for the authors below

This is a cost-effective analysis of m-health for diabetes prevention and control based on a cluster-randomized trial. The study was well designed and the protocol was well presented. I have only some minor concerns before its publication.

1. How many people use smart mobile phones in rural Bangladesh? Is there any difference between the users and non-users? Will the equipment facility be concerned in the study?

Reply: We thank the reviewer for the comment. As we mentioned in page 7 of the manuscript, the mHealth intervention is available to all individuals who have access to a mobile phone and registered to receive the messages. Receipt of messages does not require a smart phone. Respondents receive a call, and, on answering the call, listen to a voice/music message.

According to 2014 Bangladesh DHS, around 87% of rural households in Bangladesh had at least one mobile phone in 2013. In addition, in the explorative phase of the trial and for development of mHealth intervention, we completed a survey which showed that 98% of the population had access to a mobile phone. 50% of women were the main phone holders as compared to 80% of men. The findings from this survey will be reported in a forthcoming manuscript.

Regarding the equipment facility's concerns, no major technical issue has been reported regarding the mHealth intervention. Trial process evaluation monitors receipt of messages throughout the implementation period, detecting and responding to any issues with message delivery.

2. I did not find QALY in the outcomes. How do the authors think about QALY?

Reply: We agree that QALY is an important outcome to measure, but because this is a population impact trial and we don't follow the same individuals at baseline and end-line, it is not possible to estimate QALY gained. However, we have collected quality of life data at both baseline and end-line impact evaluation surveys, using EQ-5D instrument, and we will report mean quality of life score (by arm) as one of our secondary outcomes.

3. Some critical information may be re-stated in the current protocol, such as the brief timeline of the trial (eg. the time when the first cluster/patient recruited) and the strategy of informed consent (or exempted).

Reply: We thank the reviewer for their comment. We have now added the start date for the interventions' implementation (Page 6, last paragraph) and also some information on informed consent in the ethics section (page 14). We have also stated in the paper (page 7, first paragraph) that the detailed information on the clusters/villages and participants recruitment process, and trial time-line are provided in the main trial protocol (Fottrell et al. The effect of community groups and mobile phone messages on the prevention and control of diabetes in rural Bangladesh: study protocol for a three-arm cluster randomised controlled trial. *Trials*. 2016;17(1):600. and the ISRCTN trial registration <https://doi.org/10.1186/ISRCTN41083256>.)

4. I do suggest a brief discussion added following the methodology section. Some information could be discussed, such as the clinical and public expectation of the study as well as its expected strength and limitations.

Reply: We thank the reviewer for their comment. We have now added a section at the end of the discussion on page 13, which try to address the reviewer's comments, briefly explaining the potential strengths and limitation of the study.

VERSION 2 – REVIEW

REVIEWER	Sheyu Li West China Hospital, Sichuan University
REVIEW RETURNED	17-May-2018

GENERAL COMMENTS	Thanks for the careful revise of the manuscript and the response to the comments. I have no more concerns.
--

REVIEWER	Samantha Paige University of Florida; USA
REVIEW RETURNED	03-Jun-2018

GENERAL COMMENTS	Thank you for considering the comments from my initial review. I believe this protocol is suitable for publication and will greatly contribute to addressing health inequities among residents of rural Bangladesh.
---